# Radiophysiomics: Brain Tumors Classification by Machine Learning and Physiological MRI Data

**DOI:** 10.3390/cancers14102363

**Published:** 2022-05-10

**Authors:** Andreas Stadlbauer, Franz Marhold, Stefan Oberndorfer, Gertraud Heinz, Michael Buchfelder, Thomas M. Kinfe, Anke Meyer-Bäse

**Affiliations:** 1Institute of Medical Radiology, University Clinic St. Pölten, Karl Landsteiner University of Health Sciences, A-3100 St. Pölten, Austria; gertraud.heinz@stpoelten.lknoe.at; 2Department of Neurosurgery, Friedrich-Alexander University (FAU) Erlangen-Nürnberg, D-91054 Erlangen, Germany; michael.buchfelder@uk-erlangen.de (M.B.); thomasmehari.kinfe@uk-erlangen.de (T.M.K.); 3Department of Neurosurgery, University Clinic of St. Pölten, Karl Landsteiner University of Health Sciences, A-3100 St. Pölten, Austria; franz.marhold@stpoelten.lknoe.at; 4Department of Neurology, University Clinic of St. Pölten, Karl Landsteiner University of Health Sciences, A-3100 St. Pölten, Austria; stefan.oberndorfer@stpoelten.lknoe.at; 5Division of Functional Neurosurgery and Stereotaxy, Friedrich-Alexander University (FAU) Erlangen-Nürnberg, D-91054 Erlangen, Germany; 6Department of Scientific Computing, Florida State University, 400 Dirac Science Library, Tallahassee, FL 32306-4120, USA; ameyerbaese@fsu.edu

**Keywords:** brain tumors, pretreatment classification, artificial intelligence, machine learning, physiological MRI, neuro-oncology, multiclass classification

## Abstract

**Simple Summary:**

The pretreatment diagnosis of contrast-enhancing brain tumors is still challenging in clinical neuro-oncology due to their very similar appearance on conventional MRI. A precise initial characterization, however, is essential to initiate appropriate treatment management, which can substantially differ between brain tumor entities. To overcome the disadvantage of the low specificity of conventional MRI, several new neuroimaging methods have been developed and validated over the past decades. This increasing amount of diagnostic information makes a timely evaluation without computational support impossible in a clinical setting. Artificial intelligence methods such as machine learning offer new options to support clinicians. In this study, we combined nine common machine learning algorithms with a physiological MRI technique (we named this approach “radiophysiomics”) to investigate the effectiveness of the multiclass classification of contrast-enhancing brain tumors in a clinical setting. We were able to demonstrate that radiophysiomics could be helpful in the routine diagnostics of contrast-enhancing brain tumors, but further automation using deep neural networks is required.

**Abstract:**

The precise initial characterization of contrast-enhancing brain tumors has significant consequences for clinical outcomes. Various novel neuroimaging methods have been developed to increase the specificity of conventional magnetic resonance imaging (cMRI) but also the increased complexity of data analysis. Artificial intelligence offers new options to manage this challenge in clinical settings. Here, we investigated whether multiclass machine learning (ML) algorithms applied to a high-dimensional panel of radiomic features from advanced MRI (advMRI) and physiological MRI (phyMRI; thus, radiophysiomics) could reliably classify contrast-enhancing brain tumors. The recently developed phyMRI technique enables the quantitative assessment of microvascular architecture, neovascularization, oxygen metabolism, and tissue hypoxia. A training cohort of 167 patients suffering from one of the five most common brain tumor entities (glioblastoma, anaplastic glioma, meningioma, primary CNS lymphoma, or brain metastasis), combined with nine common ML algorithms, was used to develop overall 135 classifiers. Multiclass classification performance was investigated using tenfold cross-validation and an independent test cohort. Adaptive boosting and random forest in combination with advMRI and phyMRI data were superior to human reading in accuracy (0.875 vs. 0.850), precision (0.862 vs. 0.798), F-score (0.774 vs. 0.740), AUROC (0.886 vs. 0.813), and classification error (5 vs. 6). The radiologists, however, showed a higher sensitivity (0.767 vs. 0.750) and specificity (0.925 vs. 0.902). We demonstrated that ML-based radiophysiomics could be helpful in the clinical routine diagnosis of contrast-enhancing brain tumors; however, a high expenditure of time and work for data preprocessing requires the inclusion of deep neural networks.

## 1. Introduction

Cancers of the central nervous system (CNS) represent a heterogeneous group of solid neoplasms inside the skull mainly originating from the brain tissue (primary brain tumors), lymphatic tissue (primary central nervous system lymphomas, PCNSLs), or membranes that envelop the brain, also known as the meninges (meningiomas). Alternatively, the spread of cancers located in other parts of the body into the CNS leads to secondary brain tumors, also known as brain metastases. Glioblastoma (GBM WHO grade 4) is the most common and most lethal primary brain tumor in adults with a median overall survival of 14–17 months [1] and a 5-year survival rate of less than 5% [2]. Anaplastic glioma (AG or glioma WHO grade 3) is also an aggressive primary brain tumor that often affect young adults in the prime of life, causing significant disability as well as death [3]. PCNSLs account for 2–4% of all primary brain tumors [4], and the prognosis of these tumors has significantly improved in recent decades, where median survival increased from 2.5 to 26 months [5]. Meningiomas are among the most common intracranial tumors, with an estimated incidence of eight cases per 100,000 persons per year [6] with the common type (meningioma WHO grade I) as a slow-developing benign tumor [7]. Brain metastases are found in 10–30% of adult neuro-oncologic patients with cancer at another location in the body, and nearly half of these cases become clinically apparent as solitary metastases on clinical imaging [8]. All five brain tumor entities together represent by far the largest proportion of brain tumors encountered in clinical neuro-oncology.

The pretreatment diagnosis of brain tumors using conventional magnetic resonance imaging (cMRI) is still challenging in clinical neuroradiology due to their very similar appearance as hyperintense brain lesions on contrast-enhanced T1-weighted MRI surrounded by a hyperintense edema on T2-weighted MRI [9,10]. A precise and reliable initial characterization, however, is essential to initiate appropriate treatment management, which can substantially differ between the brain tumor entities [11,12]. The current standard of treatment for newly diagnosed GBMs and AGs consists of maximal possible resection of the tumor, followed by adjuvant radiotherapy and chemotherapy with temozolomide [13]. Despite tumor biopsy, PCNSLs should not undergo a total, gross resection as these brain tumors are highly responsive to steroids and high-dose methotrexate-based chemotherapy alone or in combination with whole-brain radiation therapy [14]. On the other hand, for meningiomas, surgical resection is essential in symptom-causing tumors; however, observation with close follow-up MRIs is recommended if the meningioma is small and asymptomatic [15]. Finally, stereotactic radiosurgery is considered an effective strategy in the treatment of brain metastases with the advantage of excellent local control rates with minimal invasiveness [16]. Consequently, the accurate preoperative differentiation of contrast-enhancing brain tumors is critical for individualized therapeutic decision making.

In order to overcome the disadvantage of a low specificity of cMRI in contrast-enhancing brain tumor diagnosis, a large number of new imaging methods were developed and tested over the past decades. These include, to name just a few without claiming a complete list, methods for perfusion imaging, such as arterial spin labeling (ASL) or dynamic contrast-enhanced perfusion MRI, as well as methods for metabolic or molecular imaging, such as chemical exchange saturation transfer (CEST) MRI, MR spectroscopy, and positron emission tomography (PET) using radiolabeled amino acids, which includes ^11^C-methyl-methionine (MET) or ^18^F-fluoroethyl-tyrosine (FET).

The known physiological connection between neovascularization and tissue hypoxia [17] drives our rationale for the development of a physiological MRI (phyMRI) approach for the combined quantitative characterization of microvascular architecture, neovascularization activity, and oxygen metabolism, including tissue hypoxia in CNS [18,19] and breast tumors [20,21]. Vascular architectural mapping (VAM) is an important part of the phyMRI approach. The physical basis for this MRI-based assessment of microvasculature is the different sensitivity of gradient-echo and spin-echo MRI sequences to magnetic susceptibility [22]. As a result, gradient-echo MR signals, which are commonly used for perfusion measurements in clinical routines, are dominated by larger vessel diameters starting from 20 mm, i.e., larger arterioles and venules [22], with reduced sensitivity to the smaller microvascular range. In contrast, spin-echo MR signals exhibit a peak sensitivity to the microvasculature at a vessel diameter of around 10 mm [22], including capillaries and both small arterioles and venules. Therefore, the combination of both gradient-echo and spin-echo MRI perfusion technology, as used in the phyMRI approach, allows for the examination of the entire physiological range of neovascular vessel diameters and structures. Therefore, VAM provides deeper insights into tissue microvascularity and tumor neovascularization. Additionally, the multiparametric quantitative blood-oxygenation-level-dependent (qBOLD) technique was proposed to non-invasively investigate quantitative information about tissue oxygenation metabolism and hypoxia [23]. This is also part of the phyMRI approach. Available techniques for the examination of oxygen metabolism, however, are not well-suited for in vivo investigations in humans due to their invasiveness (e.g., oxygen electrodes) or their limited availability and high costs (e.g., ^15^O positron emission tomography). This has so far hampered the implementation of such physiological measures in clinical studies on larger patient scales, as well as their use in clinical routine.

The increasing number of additional imaging parameters derived from these innovative methods generates a large amount of complex neuroimaging data. A timely evaluation of this amount of diagnostic information, which could potentially be implemented in clinical routine, is costly and hardly feasible without considerable computational support. Methods of artificial intelligence (AI), such as deep learning and traditional machine learning (ML), offer new options to support clinicians [24,25]. In particular, as shown previously for histopathology, the AI-based analysis of imaging data allows for the combined evaluation of a multitude of imaging parameters via the generation of multiparametric models [26,27]. On one hand, this makes it possible to cope with the large amounts of data, and on the other hand, it may help to increase the comparability of the obtained results as it is independent from the experience level of the evaluating clinician. Moreover, AI offers the potential to extract yet undiscovered features from routinely acquired images. Specifically, quantitative and semi-quantitative image features can be extracted from neuroimaging data, which are usually beyond human perception [28]. Hundreds of texture and histogram-based parameters can be extracted from a single data set, further increasing data volume and making it unmanageable for the clinician. The computation, identification, and extraction of image features, as well as the generation of mathematical models for characterization or prognosis prediction, is summarized under the term radiomics [29,30]. However, multiparametric ML models were predominantly used for glioma grading or the binary classification of brain tumors (e.g., GBM versus brain metastasis), and phyMRI data of microvascular architecture, neovascularization activity, or oxygen metabolism have not yet been included.

To best of our knowledge, our group was the first to use both the VAM and the qBOLD techniques in combination to simultaneously collect complementary information on tumor neovascularization and oxygen metabolism. We implemented the phyMRI approach as part of our clinical routine MRI protocol for several years. This enabled us, for the first time, to collect the physiological MRI data of neovascularization and oxygen metabolism from a sufficiently large number of brain tumor patients to justify the use of ML methods. Furthermore, the large amount of additional neuroimaging data derived from the phyMRI protocol for each patient makes the timely evaluation of this potentially valuable data impossible in clinical practice. This motivated us to evaluate the usefulness of AI technologies for this purpose. Our hypothesis was that the combining of AI technologies with high-dimensional radiomic features from phyMRI data (we named this approach “radiophysiomics”) would improve the multiclass classification of contrast-enhancing brain tumors in a clinical setting. The purpose of this first study was to investigate the effectiveness of radiophysiomics using traditional ML algorithms in multiclass classification of contrast-enhancing brain tumors. We were able to demonstrate that ML-based radiophysiomics were superior to both human reading and the ML-based classification of cMRI data for several performance parameters, including accuracy and classification error. However, the high expenditure of time and work for data processing associated with this ML-based technology requires the use of deep neural networks in future studies, as well as for the possible implementation in clinical routine.

## 2. Materials and Methods

### 2.1. Ethics

The study was approved and publicly registered by the Ethics Committee of the Lower Austrian Provincial Government (protocol code GS1-EK-4/339-2015, date of approval: 29 February 2016). The study was conducted in accordance with the guidelines of the Declaration of Helsinki. All included patients provided written informed consent prior to enrolment.

### 2.2. Patients

Patients with untreated contrast-enhancing brain tumors that were newly diagnosed between January 2016 and January 2022 were selected from a prospectively populated institutional brain MRI database. Inclusion criteria were: (i) age ≥ 18 years; (ii) histopathological confirmation of one of the following brain tumor entities: glioblastoma (GBM, WHO grade 4), anaplastic glioma (AG, WHO grade 3), primary central nervous system lymphoma (PCNSL), meningioma, or brain metastasis; (iii) no previous treatment of the brain tumor; (iv) MRI examinations with the study protocol; and (v) clinical routine MRI data evaluated by at least two board-certified radiologists in consensus.

### 2.3. MRI Data Acquisition

All MRI examinations were performed on a 3 Tesla whole-body scanner (Trio, Siemens, Erlangen, Germany) that was equipped with the standard 12-channel head coil. The MRI study protocol consisted of three parts:The conventional anatomical MRI (cMRI) protocol for clinical routine diagnosis of brain tumors included, among others, an axial fluid-attenuated inversion recovery (FLAIR) sequence as well as a high-resolution contrast-enhanced T_1_-weighted (CE T1w) sequence.The advanced MRI (advMRI) protocol for clinical routine diagnosis of brain tumors was extended by axial diffusion-weighted imaging (DWI; b values 0 and 1000 s/mm^2^) sequence and a gradient echo dynamic susceptibility contrast (GE-DSC) perfusion MRI sequence, which was performed using 60 dynamic measurements during administration of 0.1 mmol/kg-bodyweight gadoterate-meglumine (Dotarem, Guerbet, Aulnay-Sous-Bois, France).The physiological MRI (phyMRI) protocol included the innovative MRI techniques of vascular architecture mapping (VAM) [31] for the assessment of microvascular architecture and neovascularization activity, as well as the quantitative blood-oxygenation-level-dependent (qBOLD) imaging approach [19,32] for assessment of tissue oxygen metabolism and tension. The VAM approach [33,34] additionally required a spin-echo DSC (SE-DSC) perfusion MRI sequence conducted with the same parameters and contrast agent injection protocol as described for the routine GE-DSC perfusion MRI. Details of our strategy to minimize adverse effects due to differences in time to first-pass peak and contrast-agent leakage, which could significantly affect the data evaluation, were previously described [33,34]. The qBOLD approach [19,32] additionally required a multi-echo GE sequence and a multi-echo SE sequence for the mapping of the transverse relaxation rates R_2_* (=1/T_2_*) and R_2_ (=1/T_2_), respectively. All phyMRI sequences for VAM and qBOLD were carried out with identical geometric parameters (voxel size, number of slices, etc.) and slice position as used for the routine GE-DSC perfusion sequence. The phyMRI protocol required seven minutes of extra scan time in total.

### 2.4. MRI Data Processing and Calculation of MRI Biomarker Maps

Processing of advMRI and phyMRI data, as well as calculation of MRI biomarkers, was performed with custom-made MatLab (MathWorks, Natick, MA) software. Processing of the advMRI data included calculation of the apparent diffusion coefficient (ADC) maps from DWI data using the following equation:(1)$$\mathrm{ADC}=-{\ln[(S/S}_{0})/b]$$
where S_0_ is the MRI signal without diffusion-weighted sensitization (b = 0 s/mm^2^), and S represents the MRI signal measured for b = 1000 s/mm^2^. Furthermore, absolute cerebral blood volume (CBV) and flow (CBF) maps from the GE-DSC perfusion MRI data were determined via the automatic identification of arterial input functions (AIFs) [35,36].

Processing of phyMRI data for the VAM technique included a correction for remaining contrast agent extravasation as previously described [37,38]: fitting of the first bolus curves for each voxel of the GE- and SE-DSC perfusion MRI data with a previously described gamma-variate function [39], and the calculation of the ∆R_2,GE_ versus (∆R_2,SE_)^3/2^ diagram [40], the so-called vascular hysteresis loop (VHL) [33,34]. For calculation of phyMRI biomarker maps of microvascular architecture [41] including microvessel density (MVD) and vessel size index (VSI, i.e., microvessel radius), the VHL curve data and the following equations were used:(2)$$\mathrm{MVD}=\frac{Q_{\max}}{\beta }\times {\left(\frac{CBV^{2}}{4\pi^{2}\times ADC\times {\bar{R}}^{4}}\right)}^{1/3}$$
and:(3)$$\mathrm{VSI}={\left(\frac{{\mathrm{CBV}\times \mathrm{ADC}\times \beta }^{3}}{{2\pi \times Q}_{\max}^{3}}\right)}^{1/2}$$
with Qmax = max[∆R_2,GE_]/max[(∆R_2,GE_)^3/2^]. $\bar{R}$ ≈ 3.0 μm is the mean vessel lumen radius, β is a numerical constant (β = 1.6781) [41], CBV is the cerebral blood volume, and ADC is the apparent diffusion coefficient. Neovascularization activity estimated by the microvessel type indicator (MTI) was previously [33] defined as the area of the VHL curve signed with the rotational direction of the VHL curve, i.e., a clockwise VHL direction was identified with a plus sign, and a counter-clockwise VHL direction was identified with a minus sign [33]. Based on previous studies [33,42], a positive MTI value (assigned to warm colors in the MTI maps) was associated with a vascular system dominated by arterioles, whereas a negative MTI value (cool colors in the MTI maps) was associated with venule- and capillary-like vessel components. Finally, the map for the microvascular cerebral blood volume (μCBV) was calculated from the SE-DSC perfusion MRI data via a separate automatic identification of AIFs [36].

Processing of phyMRI data for the qBOLD approach required corrections for background fields of the R_2_*-mapping data [43] and for stimulated echoes of the R_2_-mapping data [44] followed by calculation of R_2_* and R_2_ maps from the multi-echo relaxometry data. For the calculation of phyMRI biomarker maps of tissue oxygen metabolism, including oxygen extraction fraction (OEF) and cerebral metabolic rate of oxygen (CMRO_2_) [32] the following equations were used:(4)$$\mathrm{OEF}=\frac{R_{2}^{*}{-R}_{2}}{\frac{4}{3}{\times \pi \times \gamma \times \Delta \chi \times \mathrm{Hct}\times B}_{0}\times \mathrm{CBV}}$$
(5)$${\mathrm{CMRO}}_{2}=\frac{C_{a}\;\times \;\mathrm{CBF}}{\frac{4}{3}{\times \pi \times \gamma \times \Delta \chi \times \mathrm{Hct}\times B}_{0}\times \mathrm{CBV}}\times \left(R_{2}^{*}-{\;R}_{2}\right)$$
where R_2_* and R_2_ are the transverse relaxation rates calculated as described above, γ (2.67502 × 10^8^ rad/s/T) is the nuclear gyromagnetic ratio; Δχ = 0.264 × 10^–6^ is the difference between the magnetic susceptibilities of fully oxygenated and fully deoxygenated haemoglobin; Hct = 0.42 × 0.85 is the microvascular hematocrit fraction, whereby the factor 0.85 stands for a correction factor of systemic Hct for small vessels, and Ca = 8.68 mmol/mL is the arterial blood oxygen content [45]. Maps of tissue oxygen tension (PO_2_) [46,47] were calculated by:(6)$${\mathrm{PO}}_{2}{=P}_{50}\times \sqrt[{h}]{\left(\frac{2}{\mathrm{OEF}}-1\right)}-\frac{{\mathrm{CMRO}}_{2}}{L}$$
where P_50_ is the hemoglobin half-saturation tension of oxygen (27 mmHg), h is the Hill coefficient of oxygen binding to hemoglobin (2.7), and L (4.4 mmol/Hg per minute) is the tissue oxygen conductivity as defined by Vafaee and Gjedde [48].

In summary, the data processing resulted in two advMRI biomarker maps for microstructural density (ADC) and macrovascular perfusion (CBV) as well as seven phyMRI biomarker maps representing microvascular perfusion (µCBV), microvascular architecture (MVD and VSI), neovascularization activity (MTI), tissue oxygen metabolism (OEF and CMRO_2_), and tissue oxygen tension (PO_2_).

### 2.5. Radiomic Feature Extraction

The overall study pipeline is shown in Figure 1. The data for cMRI (CE T1w and FLAIR), advMRI (ADC and CBV), and phyMRI (µCBV, MVD, VSI, OEF, CMRO_2_, and PO_2_) of a patient were loaded into the open-source software platform 3D Slicer (v. 4.11; https://www.slicer.org/, accessed on: 30 April 2021) and geometrically aligned. Segmentation of the tumor volume was performed on CE T1w MRI data defined as the contrast-enhancing areas [49]. Segmentation of the peritumoral edema was performed on FLAIR data defined as hyperintense areas, excluding the contrast-enhancing or necrotic portions. Regions of interest (ROIs) were manually drawn on all axial slices, showing the features for 3D segmentation by a radiologist (G.H., with 30 years of experience in neuro-oncological imaging) and confirmed by another neurosurgeon (F.M., with 15 years of experience). Disagreements were resolved by discussion until agreement. Both readers were blinded to the histopathological diagnosis of the tumor.

Grey-level intensity values of the cMRI were normalized by subtracting the mean intensity and dividing by the standard deviation with an expected resulting range [–3, 3], a mean of 0 and standard deviation of 1 in the normalized image. This procedure is also known as z-score normalization [50,51]. The grey-level discretization was carried out [52] with a bin width value of 0.1, resulting in histograms with approximately 60 bins. Biomarker maps for both advMRI and phyMRI represented quantitative imaging data with a range of physiological reasonable values. Individually adapted thresholds were applied to the biomarker maps in order to remove non-physiological values due to imaging artefacts (e.g., motion or susceptibility artefacts). Biomarker value discretization was performed with adapted bin width values in order to obtain histograms with 60–67 bins. Table 1 summarizes the value ranges and bin sizes used for the biomarker maps. Next, MRI data were resampled into a uniform voxel size of 1 × 1 × 1 mm^3^ across all patients [52].

Radiomic features were extracted with the built-in package SlicerRadiomics implemented in the 3D Slicer platform based on the Python package PyRadiomics [53]. Procedures and features were in accordance with the Imaging Biomarker Standardization Initiative (IBSI) [54]. The following features were calculated:Fourteen shape features, which represent the three-dimensional size and shape of the segmented volume of interest (VOI, i.e., contrast-enhancing tumor and peritumoral edema). These features included elongation, flatness, least and major axis length, maximum 2D diameter of column, maximum 2D diameter of row, maximum 2D diameter of slice, maximum 3D diameter, mesh volume, minor axis length, sphericity, surface area, surface volume ratio, and voxel volume.Eighteen first-order features, which represent the distribution of gray values within an image, were calculated from the histogram of voxel intensities. These features included the 10th and 90th percentile, energy, entropy, interquartile range, kurtosis, maximum, mean absolute deviation, mean, median, minimum, range, robust mean absolute deviation, root mean squared, skewness, total energy, uniformity, and variance.Seventy-five texture features, which describe relationships between neighboring voxels with similar or dissimilar values. These features included the following 6 subcategories: (i) 24 gray-level co-occurrence matrix (GLCM) features characterizing how often pairs of voxels with specific intensity levels and spatial relationships occurred in an image [55]; (ii) 14 gray-level dependence matrix (GLDM) features representing the dependency of connected voxels to a center voxel [56]; (iii) 16 gray-level run-length matrix (GLRLM) features evaluating the length of consecutive pixels with the same gray level [57]; (iv) 16 gray-level size zone matrix (GLSZM) features quantifying the number of connected voxels that share the same intensity value [58]; and (v) 5 neighboring gray-tone difference matrix (NGTDM) features assessing differences between pixel values and neighbor average gray values [59]. Schematic representation of the extraction of first-order features, GLDM features, and NGTDM features is depicted in Appendix A.

This resulted in 107 features that were extracted for both VOIs, i.e., contrast-enhancing tumor and peritumoral edema, respectively. A detailed overview and description of these radiomic features can be found elsewhere [53,60]. Mathematical formulas are described on the website of the package (https://pyradiomics.readthedocs.io, accessed on: 30 April 2021).

An edge-enhancement Laplacian of Gaussian (LoG) filter was also applied, which led to 93 additional features (18 first-order + 75 texture features) and resulted in 200 features per imaging data set and per VOI, i.e., 400 features per imaging data set in total. The numbers of features for the three MRI approaches were as follows: 800 features for cMRI data (CE T1w and FLAIR for CE tumor and edema); 1600 features for advMRI data (800 cMRI features + 400 ADC features + 400 CBV features); and 2800 features for phyMRI data (400 features × 7 biomarker maps).

### 2.6. Radiomic Feature Reduction and Selection

For radiomic feature reduction, the feature stability against perturbations in tumor segmentation was assessed. For this purpose, another coauthor (A.S., medical physicist with 22 years of experience in brain cancer imaging) manually defined ROIs in 50 randomly sampled patients. The features were extracted using the same methods as described above, and intra-class correlation coefficient (ICC) was calculated for each radiomic feature using SPSS (version 21, IBM, Chicago, IL, USA). The cMRI features with “excellent” reproducibility (ICC ≥ 0.9) were included in the further analysis [61]. Radiomic features with ICC values below this threshold were discarded from further analysis, as shown previously [62].

Radiomic feature selection was performed with the open-source software package Weka (version 3.8.5, University of Waikato, Hamilton, New Zealand). Due to the rather large number of features, this was performed in two steps following the strategy of combining the advantages of both ranking methods and learner-based methods as previously described [63]. In a first step, we applied six different attribute evaluation filters (Correlation, GainRatio, InfoGain, OneR, RefiefF, and SymmetricalUncert) in combination with the Ranker search method. Those 50 features that were top ranked in at least 3 filter rankings were selected. In a second step, we applied the learner-based feature selection method Wrapper in combination with the BestFirst search method and the respective ML algorithm, which were used for model development (we refer to the next chapter for a detailed description) to the shorten feature list. This approach enabled us to specifically select the most suitable features for each ML algorithm, i.e., an individual feature list was generated for each ML classifier. As we expected unbalanced classes due to different prevalence and well-known differences in patient numbers at our institution for the brain tumor entities, the synthetic minority oversampling technique (SMOTE) [64], which is part of the preprocessing module of WEKA, was employed in order to balance the classes. This was followed by the application of a randomize filter.

### 2.7. Model Development and Validation

Model development was also carried out with the software package WEKA. Nine commonly used ML algorithms from different families of classifiers were used for multiclass differentiation between five contrast-enhancing brain tumor entities: GBM, AG, PCNSL, meningioma, and brain metastasis, respectively, calculated using Naïve Bayes (NB), logistic regression (Log), support vector machine (SVM; with polynomial kernel) [65], k-nearest neighbors (kNN; k = 3) [66], decision tree (DT; “J48” in WEKA), multilayer perceptron (MLP; one hidden layer, number of neurons = number of features + number of classes), adaptive boosting (AdaBoost; using decision tree “J48” as classifier), random forest (RF), and bootstrap aggregating (bagging; using RandomTree as classifier). The data from the five possible combinations of MRI methods (cMRI, advMRI, phyMRI, cMRI + phyMRI, and advMRI + phyMRI) for three VOIs (CE tumor, peritumoral edema, and CE tumor + peritumoral edema) were used as training data sets, i.e., we generated and evaluated 135 different ML classification models. Short descriptions of the ML algorithms are provided in the Appendix A.

A tenfold cross-validation procedure was adopted for validation of the models. The main performance evaluation metric was the area under the receiver operating characteristic curve (AUROC) [67]. In addition, confusion matrix-derived metrics including accuracy, sensitivity (aka recall or true positive rate), specificity (aka true negative rate), precision (aka positive predictive value), and the F-score were calculated. The performance metrics for each brain tumor entity were calculated by averaging the ten different validation performances followed by the calculation of the weighted average over the five brain tumor entities.

### 2.8. Model Performance Testing and Human Reading

The four best-performing classifiers were selected for performance testing using independent test data. For this purpose, the classifiers were trained with the data of the entire training cohort. The generated models were saved and evaluated with the unseen data of an independent test cohort using metrics derived from the AUROC and confusion matrix, as described above.

At least two board-certified radiologists reviewed the advMRI data (i.e., FLAIR, CE T1w, ADC, and CBV) and other anatomical MRI data, which were part of the routine protocol, but were not included in ML procedures during clinical routine diagnosis of contrast-enhancing brain tumors. The readers had access to the clinical information of each patient. The readers recorded a final agreed diagnosis for each patient, and the most likely diagnosis was used for assessing the diagnostic performance of the human readers, which was compared with the performance of the ML models.

## 3. Results

### 3.1. Patient Characteristics

The institutional brain MRI database that was searched for this study contained a total of more than 1700 MR examinations using the study protocol in 560 brain tumor patients. From January 2016 to August 2021, a total of 167 patients (81 females; 86 males; mean age 62.0 ± 13.0 years; 21–91 years) with newly diagnosed, untreated contrast-enhancing brain tumors satisfied the inclusion criteria. These patients were selected as the training and validation cohort for the ML classifiers. The patient characteristics for this cohort were as follows:Seventy-seven patients (46%; 32 females; 45 males; mean age 63.2 ± 12.3 years; 31–84 years) had the diagnosis of a glioblastoma WHO grade 4;Seventeen patients (10%; 7 females; 10 males; mean age 49.9 ± 16.1 years; 21–73 years) had an anaplastic glioma WHO grade 3;Twenty-eight patients (17%; 18 females; 10 males; mean age 60.3 ± 13.2 years; 27–82 years) had a meningioma (15 patients WHO grade I, 12 patients WHO grade II; one patient WHO grade III);Sixteen patients (10%; 8 females; 8 males; mean age 69.8 ± 9.7 years; 55–92 years) had a PCNSL;Twenty-nine patients (17%; 16 females; 13 males; mean age 63.4 ± 9.3 years; 46–79 years) suffered from a brain metastasis that originated in twelve patients from lung cancer, in five patients from breast cancer, in four patients from a melanoma, in two patients each from esophageal or renal cancer, and in one patient each from fibrosarcoma, bladder cancer, pancreatic cancer, and colon cancer, respectively.

From September 2021 to January 2022, a total of 20 patients (11 females; 9 males; mean age 58.5 ± 15.9 years; 24–79 years) with newly diagnosed, untreated contrast-enhancing brain tumors satisfied the inclusion criteria. These patients were selected as the independent test cohort for the selected ML models. The patient characteristics were:Nine patients (45%; 4 females; 5 males; mean age 60.9 ± 18.5 years; 26–79 years) had a glioblastoma WHO grade 4;Three patients (15%; 2 females; 1 male; mean age 45.8 ± 19.8 years; 24–63 years) had an anaplastic glioma WHO grade 3;Three patients (15%; 1 female; 2 males; mean age 55.6 ± 16.2 years; 37–68 years) had a meningioma (1 patient WHO grade I and 2 patients WHO grade II);Five patients (25%; 4 females; 1 male; mean age 63.4 ± 4.3 years; 59–70 years) suffered from a brain metastasis that originated in two patients from lung cancer, and in one patient each from gastrointestinal cancer, bladder cancer, and breast cancer, respectively.

There were no patients with newly diagnosed PCNSL during this period.

### 3.2. The Selected Radiomic Features

From the 200 features that were extracted for each MRI data set and VOI, respectively, 80 features with excellent reproducibility (ICC ≥ 0.9) were selected: no shape features, 11 first-order features, 34 texture features (10 GLCM, 8 GLDM, 6 GLRLM, 9 GLSZM, 1 NGTDM), 11 LoG-filtered first-order features, and 24 LoG-filtered texture features (11 GLCM, 5 GLDM, 3 GLRLM, 4 GLSZM, 1 NGTDM). The wrapper-based classifier-specific feature selection of the top-ranked 50 features yielded 8 features (bagging) to 16 features (AdaBoost) for cMRI data, 8 features (bagging) to 19 features (random forest) for advMRI data, 12 features (decision tree) to 20 features (SVM) for phyMRI data, 14 features (Naïve Bayes) to 33 features (multilayer perceptron) for cMRI + phyMRI data, and 12 features (kNN and decision tree) to 36 features (multilayer perceptron) for advMRI + phyMRI data, respectively. A detailed overview of the numbers of the selected features per imaging data set is provided in Table 2.

### 3.3. The Top ML Classifiers in the Learning/Validation Cohort

The values for AUROC, precision, and F-score for the 135 different ML classification models are presented as heat maps in Figure 2. This figure demonstrates that the performance for the multiclass classification of the models was superior for the MRI data of CE tumor when compared to the MRI data of edema, as well as the combined data for CE tumor plus edema. Therefore, we limited our further evaluation to the ML models of CE tumors. In this subset of models, random forest and AdaBoost had the highest AUROC for all MRI data sets. However, kNN showed a higher precision and F-score compared to both AdaBoost and random forest. On the other hand, a multilayer perceptron showed a high classification performance, especially for the combined MRI datasets cMRI + phyMRI and advMRI + phyMRI, respectively. Therefore, we selected these four ML classifiers for further evaluation and testing. The performance parameters of the four classifiers were summarized and compared to those for the human reading of the learning cohort in Table 3. All selected classification models showed a higher performance for the learning data when compared to the human readers.

### 3.4. The Performance of the Selected Models in the Test Cohort

The values for AUROC, precision, F-score, and classification errors for the four selected classifiers and the five MRI data sets (cMRI; advMRI; phyMRI; cMRI + phyMRI; and advMRI + phyMRI) when applied to the test cohort are presented as heat maps in Figure 3. ML classification models using random forest as a classifier showed the highest values for AUROC. Models using AdaBoost, however, had the highest precision. The highest values for F-score and the fewest errors (5 misclassifications) showed a multilayer perceptron combined with cMRI + phyMRI data, AdaBoost combined with cMRI + phyMRI data and advMRI data, and random forest combined with phyMRI data, respectively. In general, the models using kNN showed the worst classification performance. Therefore, we selected AdaBoost and random forest for a detailed comparative analysis of the classification performance in the test cohort.

The performance parameters for the two best-performing classifiers of the five MRI data sets and those for human reading in the test cohort are summarized in Table 4. Classification models using AdaBoost in combination with both advMRI data and cMRI + phyMRI data were superior in accuracy, precision, F-score, AUROC, and classification error compared to human reading. Random forest in combination with phyMRI data also showed higher values for F-score, AUROC, and classification error compared to the performance of the radiologists, although the radiologists’ performances were better in sensitivity and specificity.

The classification results of the best-performing ML models and human reading for the 20 patients of the test cohort are summarized in Table 5. AdaBoost, in combination with advMRI data, had major problems in the classification of brain metastases: four out of five metastases were misclassified. Random forest in combination with phyMRI had problems in the differentiation between GBMs and brain metastases: in four cases, a GBM was misclassified as metastasis or vice versa. AdaBoost, in combination with cMRI + phyMRI data, also had problems in the classification of GBMs and metastasis, respectively: three metastases and one GBM were misclassified. Interestingly, all three classifier models misclassified patient number 20 (Table 5) who suffered from a meningioma but was classified as GBM by the ML models. However, the radiologists correctly classified this tumor. All MRI data of this patient are depicted in Figure 4.

The radiologists had major problems in the classification of both GBM (three cases) and metastasis (two cases). One anaplastic glioma WHO grade 3 was misclassified as PCNSL. Interestingly, three of the six cases that were misclassified by the radiologists were correctly classified by all three ML models. The MRI data of a representative patient (number 15 in Table 5) is presented in Figure 5. On the other hand, the other three cases that were misclassified by the radiologists were also misclassified by two of the three ML models. The MRI data of a representative case (patient number 7 in Table 5) is depicted in Figure 6.

## 4. Discussion

A precise and reliable characterization of contrast-enhancing brain tumors is essential for individualized therapy decisions but represents a major challenge in clinical neuro-oncology. New metabolic, molecular, or physiologic neuroimaging methods that have been developed in recent decades have proven useful in improving diagnostic performance. However, the associated increase in the amount of complex neuroimaging data to be evaluated requires the inclusion of AI methods in order to support clinicians and implement applications in clinical routine. In this study, we demonstrated that the combination of ML technologies and high-dimensional radiomic features from phyMRI data were supportive of a reliably multiclass classification of contrast-enhancing brain tumors in a clinical setting. Multiclass classification models using adaptive boosting and random forest in combination with both advMRI and phyMRI data were superior to human reading in accuracy, precision, F-score, AUROC, and— the most clinically relevant aspect—classification error in an independent test cohort of 20 consecutive patients. The radiologists, however, showed a higher sensitivity and specificity.

The vast majority of previous studies have used ML methods for the binary classification of brain tumors, particularly to differentiate between two subgroups of gliomas, e.g., low-grade vs. high-grade [68,69,70] or isocitrate dehydrogenase (IDH) wild-type vs. mutated [71,72,73,74], or to differentiate between GBMs and brain metastases [75,76,77,78,79]. Tian et al. [80] assessed the discriminative ability of texture analysis using ML to distinguish GBM from anaplastic astrocytoma WHO grade 3. They used texture features from CE T1w MRI data of 123 high-grade glioma patients (76 GBM and 47 AGs) and achieved an averaged accuracy, sensitivity, specificity, and AUROC of 0.968, 0.927, 0.989, and 0.974, respectively, using linear discriminant analysis and fivefold cross-validation. However, the authors performed no evaluation of their model with an independent test cohort. Two recent studies also used only CE T1w MRI data for the ML-based binary classification of GBMs and brain metastases [77,79]. Both studies obtained the best performance with a model using a support vector machine. Ortiz-Ramon et al. [79] only used a fivefold cross-validation scheme and achieved an averaged sensitivity, specificity and AUROC of 0.82, 0.80, and 0.896, respectively. Qian et al. [77], however, performed a performance analysis with an independent test cohort and achieved an accuracy, sensitivity, specificity, and AUROC of 0.83, 0.80, 0.87, and 0.90, respectively, with their SVM-based model. Bae et al. [78] used CE T1w and T2w MRI data corresponding to our cMRI data and found that adaptive boosting was the best-performing traditional machine learning model (accuracy, 0.829; sensitivity, 0.800; specificity, 0.875; and AUROC, 0.890) for the differentiation of GBMs and brain metastases in an independent test cohort.

Only a few studies used anatomical MRI data in combination with CBV and/or ADC data for the ML-based binary classification of brain tumors. Qin et al. [81] investigated the diagnostic performance of a histogram analysis of CBV maps combined with ML methods in the binary differentiation between GBMs and brain metastases. They found that their kNN-based model had the highest accuracy (0.95) and AUROC (0.94), and that data from the peritumoral edema were not useful for separating these two entities. The latter is in good agreement with our experience in the present study. Prof. Yamashita’s research group used cMRI data in combination with ADC maps, as well as with ADC and CBV maps, for an ML-based binary differentiation between GBMs and brain metastases [75] and GBMs and PCNSL [82], respectively. They found that ML models provided significantly higher AUROC values for the differentiation between GBMs and brain metastases (AUROC, 0.92; SVM-based model) [75], as well as for the differentiation between GBMs and PCNSL (AUROC, 0.98; extreme gradient boosting-based model) compared to radiologists (AUROC, 0.72–0.86) [82].

Just as few studies have used experimental neuroimaging techniques in combination with ML methods to classify brain tumors. Sartoretti et al. [76] evaluated the utility of CEST imaging in differentiating glial brain tumors (high- and low-grade gliomas) from brain metastases. They used a tenfold cross-validation in 48 patients, and a multilayer perceptron classifier yielded the best performance in distinguishing primary glial brain tumors from brain metastases: sensitivity, 0.813; specificity, 0.811; F-measure, 0.81; and AUROC, 0.836. Tatekawa et al. [73] used advMRI data (CE T1w, FLAIR, ADC, and CBV maps) and 3,4-dihydroxy-6-[18F]-fluoro-L-phenylalanine (FDOPA) PET images of 62 patients in combination with ML in order to classify the IDH-gene status of gliomas. The parameters for classification performance of an SVM-based ML model using a leave-one-out cross-validation strategy were as follows: accuracy, 0.76; sensitivity, 0.75; specificity, 0.82; precision, 0.78; F1-score, 0.76; and AUC, 0.81, respectively. Wiestler et al. [68] assessed the performance of ML models and advMRI, including qBOLD-based OEF, to differentiate between GBM and glioma WHO grade 2/3. A random forest ML classifier yielded a fivefold cross-validated AUROC of 0.944, with 34 of 37 patients correctly classified (accuracy 91.8%). The patient numbers in these studies that used experimental neuroimaging techniques in combination with ML methods was expectedly low (37–62 patients). We were able to include 167 patients for training/validation and 20 consecutive patients for independent testing because we performed our phyMRI protocol in clinical routine diagnosis over several years.

The performance parameters of the above-mentioned studies that used cMRI or advMRI data, respectively, are comparable to or higher than those obtained in our study. One reason for this is certainly that the studies performed only a binary classification. This requires patient selection prior to classification and with only two possible classes, and the probability and possibility of error is lower compared to the classification of the five classes. The multiclass approach, on the other hand, is closer to clinical reality. We selected the five most common brain tumor entities showing contrast enhancement for our study. However, this also represents a limitation of our study because non-tumorous processes with contrast-enhancement, such as brain abscesses, or less common brain tumors, such as ependymomas, were excluded. For these pathologies, the patient numbers (<5) were too low for machine learning.

There are few studies on the multiclass classification of brain tumors using ML. Zacharaki et al. [66] developed an ML-based classification method combining cMRI and CBV data and used it for the differential diagnosis of brain tumors (24 brain metastases, 4 meningiomas, 22 gliomas WHO grade 2, 18 gliomas WHO grade 3, 34 GBMs). The highest average multiclass classification accuracy assessed by leave-one-out cross-validation was achieved using voting feature intervals (VFI; 76.3%) followed by kNN (75.3%) and naïve bayes (74.2%). Regarding brain tumor entity, metastasis (91.7%) and glioma WHO grade 2 (90.9%) showed the highest classification accuracy, whereas the classification accuracy for GBM was reduced (29.4% were classified as glioma WHO grade 3 and 29.4% as metastasis). The lowest classification rate in the multiclass problem was found for the glioma WHO grade 3; 44.4% were classified as glioma WHO grade 2 and 11.1% as GBM or metastasis. Swinburne et al. [83] finally investigated whether the ML evaluation of advMRI data of 26 patients can reliably differentiate between GBMs (*n* = 9), brain metastases (*n* = 9), and PCNSL (*n* = 8). A multilayer perceptron model discriminated between the three pathologic classes with a maximum accuracy of 69.2% using leave-one-out cross-validation. However, comparability to our results is limited, because they included fewer tumor entities and smaller patient numbers. Furthermore, no evaluation with an independent test cohort was carried out in these studies.

Here, we performed an ML-based multiclass classification of the five most common contrast-enhancing brain tumor entities using advMRI and phyMRI training data from 167 patients. Applications in an independent test cohort revealed that AdaBoost showed the highest values for accuracy (0.875), precision (0.862), and F-score (0.774), and random forest showed the highest values for AUROC (0.886). Both ML algorithms were superior in these performance parameters and showed fewer classification errors compared to human readings. However, the performance of the radiologists was better in sensitivity and specificity. Our findings are in good agreement with Payabvash et al. [65], who found that, for the multiclass differentiation of the five most common posterior fossa tumors, random forest combined with advMRI data yielded an averaged AUROC of 0.873 in an independent test cohort, which was superior compared to human readings (AUROC of 0.799–0.834).

One of the most important advantages of ML-based classification over human reading was the extraction and evaluation of quantitative and semi-quantitative image features from routinely acquired neuroimaging data, which are usually beyond human perception. In this context, it is interesting that all shape features that are most comparable to the qualitative assessment made by radiologists were excluded during feature selection. Shape features omit tumor heterogeneity and provide only morphologic parameters; therefore they are limited in their discriminative power. Texture analysis represents an efficient technique to quantify and characterize voxel-intensity distribution/heterogeneity on variable images. The derived high-ranked texture parameters determine an important textural characteristics of tumor heterogeneity and are more discriminative than the shape features of a differential diagnosis. The results confirm the fact that there is a very low probability of extracting insufficient textural information that is known for other cancers, such as breast and prostate. The LoG filter computes the average gradient direction in the vicinity of a point and provides neighborhood information by considering partial derivatives, which also prove to be beneficial.

We would like to address some limitations of our present study. This study included a relatively small number of patients for both the training/validation cohort (167 patients) and the test cohort (20 patients). Furthermore, all data were acquired at a single site with a single MRI scanner, which did not account for possible variations in examination protocols as well as different magnetic field strengths and MR scanner setups. However, physiological MR imaging, including oxygen metabolism, microvascular architecture, and neovascularization is still very rarely used in clinical routines, but this could change in the future with an increasing availability of the necessary data post-processing software. We also decided to perform traditional ML. The individual steps of the AI-based evaluation, i.e., data pre-processing, feature extraction and classification, are more transparent and easier to understand compared to deep learning approaches. Traditional ML models show a high stability and reproducibility due to their low complexity. However, all steps for data pre-processing and feature extraction as well as the application and tuning of ML classification algorithms have to be manually carried out. This is very time-consuming and labor-intensive, reducing compatibility with clinical routines as well as the number of patients that can be enrolled. Therefore, the implementation of deep neural networks for the classification of brain tumors as well as for therapy monitoring and recurrence detection in combination with physiological MRI data is the next logical step. Deep learning architectures with modern modules work very effectively and quickly, but place high demands on computer hardware, especially the graphics processing unit (GPU). This particularly applies to the processing of multi-parametric MRI data, such as 11 different MRI data sets for each patient in our study. At the time that this study was initiated, we did not have access to the appropriate hardware, which has since changed. Therefore, our future work will focus on the implementation of convolutional neural networks for brain tumor classification as well as for combined architectures of convolutional and recurrent neural networks in order to implement therapy monitoring and recurrence detection in combination with phyMRI data. The goal is to develop a clinical decision support system, as has already been described for other clinical issues [84,85,86].

## 5. Conclusions

In this paper, we introduced a novel concept, radiophysiomics, integrated with ML methods for an effective classification of contrast-enhancing brain tumors in a clinical setting. In contrast to prior studies, we used a wealth of quantitative parameters extracted from phyMRI data, including microvascular architecture, neovascularization activity, tissue oxygen metabolism and hypoxia. We determined based on a feature ranking as the most relevant discriminators a total number of up to 36 features per lesion, and employed nine robust ML algorithms. The best-performing ML models, adaptive boosting and random forest, outperformed radiologists in the multiclass classification of contrast-enhancing brain tumors when combined with quantitative physiological MRI data in the accuracy (0.875 vs. 0.850), precision (0.862 vs. 0.798), F-score (0.774 vs. 0.740), AUROC (0.886 vs. 0.813), and classification error (5 vs. 6). The radiologists, however, showed a higher sensitivity (0.767 vs. 0.750) and specificity (0.925 vs. 0.902). The radiophysiomics method provides insights into the tumor phenotype based on the radiomics signature with important diagnostic implications that have not yet been investigated. We demonstrated that ML-based radiophysiomics could be helpful in clinical routine diagnosis; however, a high expenditure of time and work requires the inclusion of deep neural networks. Furthermore, it is important to note that the clinical appliance has to be chosen by medical professionals since the existing differences between the real image and the image generated by the proposed system could be substantial in the medical field.

## Figures and Tables

**Figure 1 cancers-14-02363-f001:**
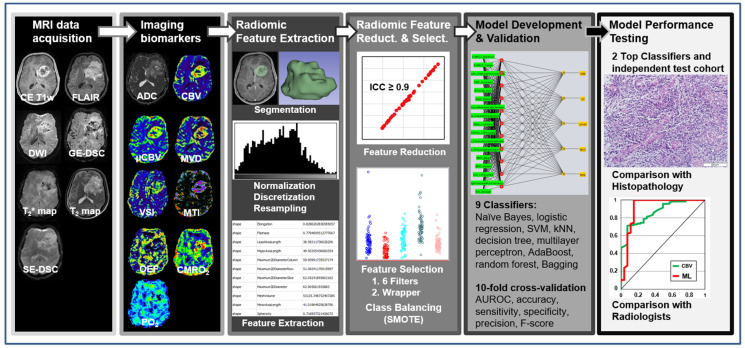
General diagram of the proposed radiomics and radiophysiomics approach, showing the major steps: MRI data acquisition; calculation of imaging biomarkers; extraction of radiomic features, including tumor and edema segmentation, data filtering, and feature extraction; reduction in and selection of the most relevant features; development and validation of ML-based classification models; and testing the performance of the best-performing classifiers.

**Figure 2 cancers-14-02363-f002:**
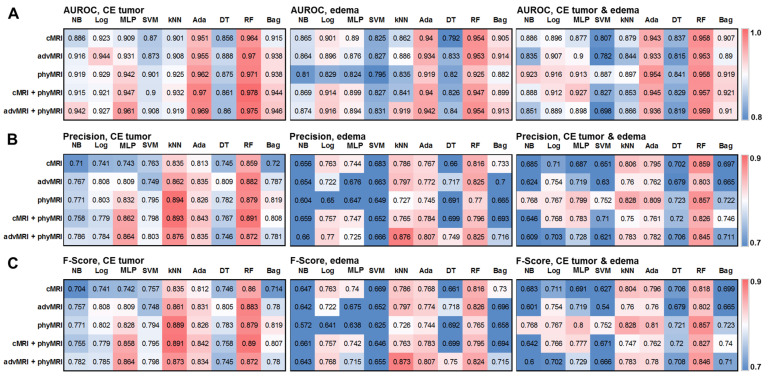
Heatmaps depicting (**A**) the area under the receiver operating characteristic (AUROC) curve, (**B**) precision, and (**C**) F-score for the five MRI data set combinations (conventional MRI (cMRI), advanced MRI (advMRI), physiological MRI (phyMRI), cMRI combined with phyMRI data, and advMRI combined with phyMRI data) and the nine ML algorithms in contrast-enhancing (CE) brain tumor (left), peritumoral edema (middle), and the combined area of CE tumor and edema (right), respectively. The color codes are listed on the far right. NB = naïve Bayes, Log = logistic regression, MLP = multilayer perceptron, SVM = support vector machine, kNN = k-nearest neighbors, Ada = adaptive boosting (AdaBoost), DT = decision tree, RF = random forest, and Bag = bootstrap aggregating (bagging).

**Figure 3 cancers-14-02363-f003:**
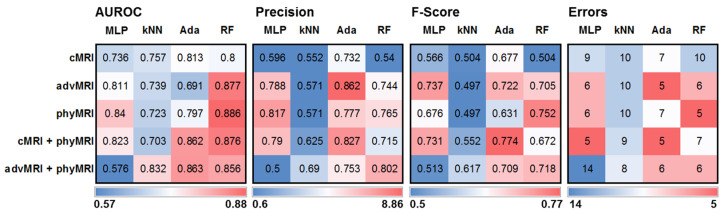
Heatmaps depicting (from left to right) the area under the receiver operating characteristic (AUROC) curve, precision, F-score, and classification error for the five MRI data set combinations in CE brain tumors and the four ML algorithms showing the best performance in training/validation, applied to independent training cohort. MLP = multilayer perceptron, kNN = k-nearest neighbors, Ada = adaptive boosting (AdaBoost), and RF = random forest; cMRI = conventional MRI, advMRI = advanced MRI, phyMRI = physiological MRI, cMRI + phyMRI = combination of cMRI and phyMRI data, and advMRI + phyMRI = combination of advMRI and phyMRI data, respectively. The color codes are below the parts of the figure.

**Figure 4 cancers-14-02363-f004:**
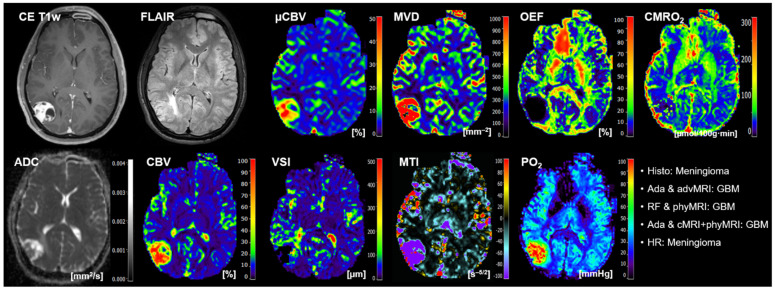
Representative case of a patient (number 20 in Table 5) suffering from a meningioma that was misclassified as GBM by all three best-performing ML classifiers but correctly classified by the radiologists. Contrast-enhanced (CE) T1w and FLAIR MRI data were conventional MRI (cMRI) data; cMRI data combined with the quantitative maps of apparent diffusion coefficient (ADC) and cerebral blood volume (CBV) were advanced MRI (advMRI) data; physiological MRI (phyMRI) data included the quantitative maps of microvascular cerebral blood volume (μCBV), microvessel density (MVD) microvessel radius (VSI), microvessel type indicator (MTI), oxygen extraction fraction (OEF), cerebral metabolic rate of oxygen (CMRO_2_), and tissue oxygen tension (PO_2_), respectively.

**Figure 5 cancers-14-02363-f005:**
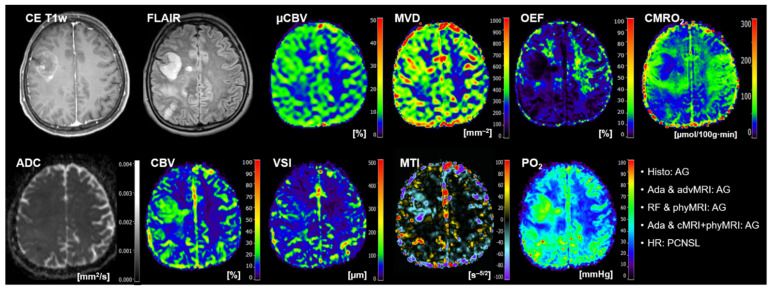
Representative case of a patient (number 15 in Table 5) suffering from an anaplastic glioma (AG, WHO grade 3) who was misclassified as primary CNC lymphoma (PCNSL) by the radiologists but correctly classified by all three best-performing ML classifiers. Contrast-enhanced (CE) T1w and FLAIR MRI data were conventional MRI (cMRI) data; cMRI data combined with the quantitative maps of apparent diffusion coefficient (ADC) and cerebral blood volume (CBV) were advanced MRI (advMRI) data; physiological MRI (phyMRI) data included the quantitative maps of microvascular cerebral blood volume (μCBV), microvessel density (MVD) microvessel radius (VSI), microvessel type indicator (MTI), oxygen extraction fraction (OEF), cerebral metabolic rate of oxygen (CMRO_2_), and tissue oxygen tension (PO_2_), respectively.

**Figure 6 cancers-14-02363-f006:**
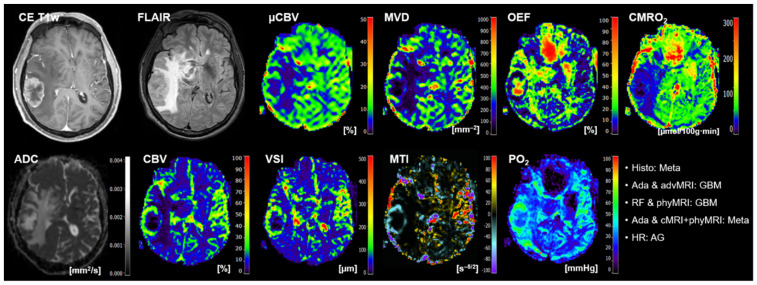
Representative case of a patient (number 7 in Table 5) suffering from a brain metastasis that was misclassified by the radiologists and by two of the three ML models. Contrast-enhanced (CE) T1w and FLAIR MRI data were conventional MRI (cMRI) data; cMRI data combined with the quantitative maps of apparent diffusion coefficient (ADC) and cerebral blood volume (CBV) were advanced MRI (advMRI) data. Physiological MRI (phyMRI) data included the quantitative maps of microvascular cerebral blood volume (μCBV), microvessel density (MVD) microvessel radius (VSI), microvessel type indicator (MTI), oxygen extraction fraction (OEF), cerebral metabolic rate of oxygen (CMRO_2_), and tissue oxygen tension (PO_2_), respectively.

**Table 1 cancers-14-02363-t001:** Value ranges, bin sizes, and bin numbers for discretization of biomarker maps.

Value	advMRI Maps		phyMRI Maps						
	ADC	CBV	µCBV	MVD	VSI	MTI	OEF	CMRO_2_	PO_2_
range	[0, 3]	[0, 100]	[0.30]	[0, 2000]	[0, 500]	[−1000, 1000]	[0, 100]	[0, 1000]	[0, 200]
unit	mm^2^/s	%	%	mm^−2^	µm	s^−2/5^	%	µmol/100 g × min	mmHg
bin size	0.05	1.5	0.5	30	8	30	1.5	15	3
bins	60	67	60	67	63	67	67	67	67

advMRI = biomarker maps for advanced conventional MRI; phyMRI = physiological MRI; ADC = apparent diffusion coefficient; CBV = cerebral blood volume; μCBV = microvascular cerebral blood volume; MVD = microvessel density; VSI = vessel size index; MTI = microvessel type indicator; OEF = oxygen extraction fraction; CMRO_2_ = cerebral metabolic rate of oxygen; and PO_2_ = tissue oxygen tension.

**Table 2 cancers-14-02363-t002:** Number of the selected radiomic features for the ML algorithms and the MRI data sets.

Caption		NB	Log	MP	SVM	kNN	Ada	DT	RF	Bag
cMRI	CE T1w	9	21	16	23	10	15	11	14	8
	FLAIR	1	1	0	1	1	1	0	1	0
	total	10	22	16	24	11	16	11	15	8
advMRI	CE T1w	10	6	6	10	4	4	4	7	5
	FLAIR	0	0	0	0	0	0	0	0	0
	ADC	2	1	2	2	1	1	1	1	1
	CBV	5	5	7	4	9	4	3	11	2
	total	17	12	15	16	14	9	8	19	8
phyMRI	CMRO_2_	3	2	3	4	1	3	2	2	1
	OEF	2	2	2	1	0	2	0	2	3
	PO_2_	1	1	0	1	0	2	0	1	1
	MIT	2	2	2	3	6	2	2	4	3
	µCBV	1	2	2	3	0	0	0	3	0
	MVD	5	3	1	4	3	4	5	2	2
	VSI	5	3	3	4	3	2	3	3	3
	total	19	15	13	20	13	15	12	17	13
cMRI + phyMRI	CE T1w	5	11	20	16	11	10	9	9	6
	FLAIR	0	0	0	0	0	0	0	0	0
	CMRO_2_	2	2	2	3	3	1	1	1	1
	OEF	2	1	0	1	1	1	1	2	1
	PO_2_	0	0	0	0	0	0	0	0	0
	MIT	1	2	5	5	3	1	1	3	3
	µCBV	1	2	1	1	1	0	0	0	2
	MVD	0	1	1	2	2	3	3	2	0
	VSI	3	2	4	3	4	3	3	5	3
	total	14	21	33	31	25	19	18	22	16
advMRI + phyMRI	CE T1w	8	8	12	13	5	4	6	6	9
	FLAIR	0	0	0	0	0	0	0	0	0
	ADC	0	0	0	0	0	0	0	0	0
	CBV	8	4	13	8	3	6	3	7	8
	CMRO_2_	1	0	1	1	0	0	1	1	1
	OEF	1	0	0	1	0	1	0	0	0
	PO_2_	0	0	1	0	0	0	0	0	0
	MIT	1	2	3	4	2	1	1	1	1
	µCBV	1	1	1	0	1	0	0	0	1
	MVD	1	0	1	1	0	0	0	0	0
	VSI	3	3	4	4	1	1	1	2	2
	total	24	18	36	32	12	13	12	17	22

advMRI = biomarker maps for advanced conventional MRI; phyMRI = physiological MRI; ADC = apparent diffusion coefficient; CBV = cerebral blood volume; μCBV = microvascular cerebral blood volume; MVD = microvessel density; VSI = vessel size index; MTI = microvessel type indicator; OEF = oxygen extraction fraction; CMRO_2_ = cerebral metabolic rate of oxygen; and PO_2_ = tissue oxygen tension; NB = naïve Bayes; Log = logistic regression; MP = multilayer perceptron; SVM = support vector machine; kNN = k-nearest neighbors; Ada = adaptive boosting; DT = decision tree; RF = random forest, Bag = bagging.

**Table 3 cancers-14-02363-t003:** Diagnostic performance of the four selected ML classifiers and the human readers in the learning/validation cohort.

Caption		Accuracy	Sensitivity	Specificity	Precision	F-Score	AUROC
RF	cMRI	0.944	0.867	0.965	0.859	0.860	0.964
	advMRI	0.953	0.889	0.970	0.882	0.883	0.97
	phyMRI	0.951	0.887	0.970	0.879	0.879	0.971
	cMRI + phyMRI	0.956	0.897	0.973	0.891	0.890	0.978
	advMRI + phyMRI	0.948	0.878	0.968	0.872	0.872	0.975
Ada	cMRI	0.924	0.817	0.952	0.813	0.812	0.951
	advMRI	0.931	0.837	0.957	0.835	0.831	0.955
	phyMRI	0.929	0.831	0.955	0.826	0.826	0.962
	cMRI + phyMRI	0.936	0.847	0.960	0.843	0.842	0.97
	advMRI + phyMRI	0.934	0.842	0.959	0.835	0.834	0.969
kNN	cMRI	0.934	0.842	0.959	0.835	0.835	0.901
	advMRI	0.944	0.867	0.965	0.862	0.861	0.908
	phyMRI	0.956	0.898	0.973	0.894	0.889	0.925
	cMRI + phyMRI	0.956	0.897	0.973	0.893	0.891	0.932
	advMRI + phyMRI	0.949	0.881	0.969	0.876	0.873	0.919
MP	cMRI	0.897	0.748	0.935	0.743	0.742	0.909
	advMRI	0.923	0.814	0.952	0.809	0.809	0.931
	phyMRI	0.931	0.834	0.957	0.832	0.828	0.942
	cMRI + phyMRI	0.943	0.864	0.965	0.862	0.858	0.947
	advMRI + phyMRI	0.945	0.869	0.965	0.864	0.864	0.961
Human Reading		0.846	0.739	0.927	0.767	0.708	0.808

cMRI = conventional MRI; advMRI = biomarker maps for advanced MRI; phyMRI = physiological MRI; RF = random forest; Ada = adaptive boosting; kNN = k-nearest neighbors; MP = multilayer perceptron; AUROC = area under the receiver operator curve.

**Table 4 cancers-14-02363-t004:** Performance parameters for the top ML classifiers of the five MRI data sets and human reading of the test cohort.

Caption		Accuracy	Sensitivity	Specificity	Precision	F-Score	AUROC	Errors
Ada	cMRI	0.815	0.650	0.852	0.732	0.677	0.813	7
Ada	advMRI	0.853	0.750	0.836	0.862	0.722	0.691	5
RF	phyMRI	0.830	0.750	0.844	0.765	0.752	0.886	5
Ada	cMRI + phyMRI	0.875	0.750	0.902	0.827	0.774	0.862	5
RF	advMRI + phyMRI	0.843	0.700	0.860	0.802	0.718	0.863	6
Human Reading		0.850	0.767	0.925	0.798	0.740	0.813	6

cMRI = conventional MRI; advMRI = advanced MRI; phyMRI = physiological MRI; RF = random forest; Ada = adaptive boosting; AUROC = weighted averaged area under the receiver operator curve.

**Table 5 cancers-14-02363-t005:** Classification results of the best-performing ML models and human reading for the test cohort.

ID	Histology	AdaBoost AdvMRI	RF PhyMRI	AdaBoost cMRI + PhyMRI	Human Reading
1	Meta	Meta	Meta	Meta	Meta
2	MNG	MNG	MNG	MNG	MNG
3	GBM	GBM	Meta (error)	GBM	GBM
4	GBM	GBM	GBM	GBM	GBM
5	GBM	GBM	GBM	GBM	PCNSL (error)
6	MNG	MNG	MNG	MNG	MNG
7	Meta	GBM (error)	GBM (error)	Meta	AG (error)
8	Meta	GBM (error)	GBM (error)	GBM (error)	Meta
9	GBM	GBM	GBM	GBM	GBM
10	GBM	GBM	GBM	GBM	GBM
11	GBM	GBM	GBM	GBM	AG (error)
12	GBM	GBM	Meta (error)	Meta (error)	Meta (error)
13	AG	AG	AG	AG	AG
14	GBM	GBM	GBM	GBM	GBM
15	AG	AG	AG	AG	PCNSL (error)
16	AG	AG	AG	AG	AG
17	Meta	GBM (error)	Meta	PCNSL (error)	GBM (error)
18	Meta	PCNSL (error)	Meta	PCNSL (error)	Meta
19	GBM	GBM	GBM	GBM	GBM
20	MNG	GBM (error)	GBM (error)	GBM (error)	MNG

cMRI = conventional MRI; advMRI = biomarker maps for advanced MRI; phyMRI = physiological MRI; RF = random forest; Ada = adaptive boosting; GBM = glioblastoma; AG = anaplastic glioma WHO grade 3; MNG = meningioma; PCNSL = primary central nervous system lymphoma; Meta = metastasis.

## Data Availability

Data available on request due to privacy and ethical restrictions.

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
