# Peer review of "Radiophysiomics: Brain Tumors Classification by Machine Learning and Physiological MRI Data"

_cancers, 2022, doi:10.3390/cancers14102363_

Round 1
Reviewer 1 Report
- Title is long and unclear. Please update the “Title”.
- In keywords, the first term should be “Capitalized”.
- Please do not use lumping referencing, for example: [18-23], [24-26].
- Literature review with enough explanation is needed in the introduction or as a new separated section.
- Use “×” instead of “.” In the equations.
- In Eq. (5), explain what “*” symbol means.
- All of the variables after equations should be explained.
- Please determine the drawback and the advantages of your research.
- Future works should be also added to the Conclusions.
- Please, include the following references in the related part of your work: (2021). Brain tumor diagnosis based on metaheuristics and deep learning. International Journal of Imaging Systems and Technology, 31(2), 657-669. (2021). Deep learning in histopathology: the path to the clinic. Nature medicine, 27(5), 775-784.; (2021). Breast Cancer Diagnosis by Convolutional Neural Network and Advanced Thermal Exchange Optimization Algorithm. Computational and Mathematical Methods in Medicine, 2021. DOI: 10.1155/2021/5595180; (2020). Measuring domain shift for deep learning in histopathology. IEEE journal of biomedical and health informatics, 25(2), 325-336; (2017). Analysis of histopathology images: From traditional machine learning to deep learning. In Biomedical Texture Analysis (pp. 281-314). Academic Press.
- The novelty of the study should be given in introduction.
- The motivation of the study should be given in introduction.
- Some important quantitative results should be given at the Abstract and Conclusion sections.
Author Response
Responses to Comments of Reviewer 1
Comment 1: Title is long and unclear. Please update the “Title”.
Response 1: We thank the reviewer for this comment. We shortened the title accordingly.
Comment 2: In keywords, the first term should be “Capitalized”.
Response 2: We thank the reviewer for pointing this out. We capitalized the first term of the keywords.
Comment 3: Please do not use lumping referencing, for example: [18-23], [24-26].
Response 3: We agree with the reviewer. We reduced lumping referencing.
Comment 4: Literature review with enough explanation is needed in the introduction or as a new separated section.
Response 4: We thank the reviewer for this important comment and agree with the reviewer that more extensive explanation of the methodology is required in the Introduction. We include a new paragraph into the Introduction (page 3, line 111-131). The literature about application of machine learning for MRI-based classification of brain tumors is extensively discussed in the discussion section.
Comment 5: Use “×” instead of “.” In the equations.
Response 5: We thank the reviewer for this important comment. We are sure that the equations are now more readable.
Comment 6: In Eq. (5), explain what “*” symbol means.
Response 6: R2* is the reciprocal of T2*, the "observed" or "effective" transverse relaxation time. T2, however, is the "natural" or "true" transverse relaxation time of the tissue. T2 is defined as a time constant for the decay of transverse magnetization arising from natural interactions at the atomic or molecular levels. It is a tissue-specific constant.
Comment 7: All of the variables after equations should be explained.
Response 7: We fully agree with the reviewer and added explanations for all variables (page 5-6, line 229-269).
Comment 8: Please determine the drawback and the advantages of your research.
Response 8: We thank the reviewer for this comment. We included this information in the discussion (page 18-19, line 716-743).
Comment 9: Future works should be also added to the Conclusions.
Response 9: We agree with the reviewer and added information about future work in the conclusions (page 19, line 760-763).
Comment 10: Please, include the following references in the related part of your work: (2021). Brain tumor diagnosis based on metaheuristics and deep learning. International Journal of Imaging Systems and Technology, 31(2), 657-669. (2021). Deep learning in histopathology: the path to the clinic. Nature medicine, 27(5), 775-784.; (2021). Breast Cancer Diagnosis by Convolutional Neural Network and Advanced Thermal Exchange Optimization Algorithm. Computational and Mathematical Methods in Medicine, 2021. DOI: 10.1155/2021/5595180; (2020). Measuring domain shift for deep learning in histopathology. IEEE journal of biomedical and health informatics, 25(2), 325-336; (2017). Analysis of histopathology images: From traditional machine learning to deep learning. In Biomedical Texture Analysis (pp. 281-314). Academic Press.
Response 10: We included the references in our work.
Comment 11: The novelty of the study should be given in introduction.
Response 11: We thank the reviewer for this important comment. We included information about novelty of our study in the Introduction (page 4, line154-173).
Comment 12: The motivation of the study should be given in introduction.
Response 12: We thank the reviewer for this important comment. We included information about our motivation for of study in the Introduction (page 4, line154-173).
Comment 13: Some important quantitative results should be given at the Abstract and Conclusion sections.
Response 13: We fully agree with the reviewer and added quantitative results in the Abstract and Conclusion section (page 19, line 754-756).

Reviewer 2 Report
In this paper, authors investigated whether multiclass machine learning (ML) algorithms applied to a high-dimensional panel of radiomic features from advanced MRI (advMRI) and physiological MRI (phyMRI; thus Radiophysiomics) could reliably classify contrast-enhancing brain tumors. This recently developed phyMRI technique enables quantitative assessment of microvascular architecture, neovascularization, oxygen metabolism, and tissue hypoxia. The following review comments are recommended, and the authors are invited to explain and modify.
Comment: Novelty is confusing. A highlight is required. The main contributions of the manuscript are not clear. The main contributions of the article must be very clear and would be better if summarize them into 3-4 points at the end of the introduction.
Comment: What is the logic to use the synthetic minority oversampling technique (SMOTE) to deal unbalanced classes? It can be also solved via a weight loss function.
Comment: “Studies on multiclass classification of brain tumors using ML are rare”, this is a vague claim.
Comments: The design of networks and approach are from my point of view outdate, nowadays completely replaceable by deep learning architectures with modern modules or a fully connected transformer. Moreover, the new approaches work very effectively and quickly.
Comment: Nothing is mentioned about the implementation challenges.
Comment: Discuss the stability of the system in terms of complexity.
Comment: The following clinical decision support systems using AI (ML/DL), and medical imaging must be included to improve the quality of the paper.
A Lightweight Convolutional Neural Network Model for Liver Segmentation in Medical Diagnosis.
SVseg: Stacked Sparse Autoencoder-Based Patch Classification Modeling for Vertebrae Segmentation.
Comment: Moreover, it should be noticed that the clinical appliance has to be decided by medicals since the existing differences between the real image and the one generated by the proposed system could be substantial in the medical field.
Comment: Could you please check your references carefully? All references must be complete before the acceptance of a manuscript.
Author Response
Responses to Comments of Reviewer 2
Comment 1: Novelty is confusing. A highlight is required. The main contributions of the manuscript are not clear. The main contributions of the article must be very clear and would be better if summarize them into 3-4 points at the end of the introduction.
Response 1: We fully agree with the reviewer. We included information about novelty of our study, our motivation for this study, and highlights in the Introduction (page 4, line154-173).
Comment 2: What is the logic to use the synthetic minority oversampling technique (SMOTE) to deal unbalanced classes? It can be also solved via a weight loss function.
Response 2: We thank the reviewer for this comment. SMOTE was successfully applied in several previous studies describing machine learning based brain tumor classification using multi-parametric MRI data (doi: 10.18632/oncotarget.18001; doi: 10.1148/radiol.2016161382; doi: 10.1002/nbm.3781; doi: 10.1016/j.ejrad.2020.108892; doi: 10.3389/fonc.2020.00071). Furthermore, SMOTE was also supported in ‘WEKA-Preprocess’ module. We included this information into the methods section (page 9, line 384).
Comment 3: “Studies on multiclass classification of brain tumors using ML are rare”, this is a vague claim.
Response 3: We agree with the reviewer. We rephrased the sentence (page 17, line 671).
Comments 4: The design of networks and approach are from my point of view outdate, nowadays completely replaceable by deep learning architectures with modern modules or a fully connected transformer. Moreover, the new approaches work very effectively and quickly.
Response 4: We agree fully with the reviewer. We discussed this issue in the discussion (page 18-19, line 727-743).
Comment 5: Nothing is mentioned about the implementation challenges.
Response 5: We discussed the challenges that are associated with the implementation of ML algorithms in the discussion (page 18-19, line 716-743).
Comment 6: Discuss the stability of the system in terms of complexity.
Response 6: We discussed the stability and complexity of the ML algorithms in the discussion (page 18-19, line 727-729).
Comment 7: The following clinical decision support systems using AI (ML/DL), and medical imaging must be included to improve the quality of the paper.
A Lightweight Convolutional Neural Network Model for Liver Segmentation in Medical Diagnosis.
SVseg: Stacked Sparse Autoencoder-Based Patch Classification Modeling for Vertebrae Segmentation.
Response 7: We included the references in our manuscript (page 19, line 743).
Comment 8: Moreover, it should be noticed that the clinical appliance has to be decided by medicals since the existing differences between the real image and the one generated by the proposed system could be substantial in the medical field.
Response 8: The reviewer is absolutely right with this important comment. We included this information in the conclusion (page 19, line 761-763).
Comment 9: Could you please check your references carefully? All references must be complete before the acceptance of a manuscript.
Response 9: We thank the reviewer for this valuable comment. We checked and corrected our references.

Round 2
Reviewer 1 Report
The authors resolved all of my concerns.
Reviewer 2 Report
The authors have answered my questions satisfactorily.